# Perinatal Depressive Symptoms among Pregnant Employees in Taiwan

**DOI:** 10.3390/ijerph20043354

**Published:** 2023-02-14

**Authors:** Su-Ying Tsai

**Affiliations:** Department of Health Management, I-Shou University, Kaohsiung 82445, Taiwan; sytsai@isu.edu.tw

**Keywords:** pregnant employees, job strain, social support, sleep problems, perinatal depressive symptoms

## Abstract

This was a longitudinal study of perinatal depressive symptoms among females employed in a large electronics manufacturer in Taiwan, conducted from August 2015 through October 2016. We used questionnaires to collect data on perceived job strain, social support, and the Edinburgh Postnatal Depression Scale (EPDS) scores at three perinatal time-points (pregnancy, delivery, and return to the workplace). Of the 153 employees who agreed to participate, 82 completed the three stages. The prevalence of perinatal depressive symptoms for the three stages was 13.7%, 16.8%, and 15.9%, respectively. The incidence at 3 weeks after childbirth and 1 month after returning to the workplace was 11.0% and 6.8%, respectively. During the third trimester of pregnancy, sleep problems (odds ratio [OR] = 6.2, 95% confidence Interval [95% CI] = 2.1–19.3), perceived job strain (OR = 4.4, 95% CI = 1.5–14.3), and lack of support from family or friends (OR = 7.0, 95% CI = 1.3–40.8) were significant risk factors. Sleep problems (OR = 6.0, 95% CI = 1.7–23.5) and lack of support from family or friends (OR = 27.6, 95% CI = 4.1–322.3) were associated with an increased risk of perinatal depressive symptoms at 3 weeks after childbirth. After returning to the workplace, perceived job strain (OR = 18.2, 95% CI = 2.2–435.7) was a significant risk factor. These findings could provide insight about early symptom detection, and more studies to clarify the association would be worthwhile.

## 1. Introduction

The population of women employed during their childbearing and child-rearing years continues to grow [1]. Women are much more likely to be diagnosed with depression during the first year after childbirth, as compared to other times [2,3]. According to a World Health Organization survey [4], 10% to 15% of women in developed countries and 20% to 40% of women in developing countries experience depression during pregnancy or after childbirth. A longitudinal study [5] at different time points during pregnancy and in the postpartum period showed that the period prevalence depression was 12.4% in pregnancy and 9.6% in the postpartum period, and the cumulative incidence of depression in pregnancy and in the postpartum period was 2.2% and 6.8%, respectively. Another study in Taiwan [6] revealed that the prevalence of perinatal depression was 13.2%, and the incidence of depression was 4.7% (16th week) and 3.4% (28th week), and 5.1% during the postpartum period. Depression during pregnancy is an important health problem because depressive symptoms not only lead to serious psychologic problems in women but also have a significant effect on the family [7,8,9]. Postpartum depression threatens not only the health of mothers but also the health of infants, and infants of depressed mothers may have developmental disorders, reduced cognitive function, as well as problems pertaining to social communication with their parents and peers [10,11,12].

In general, most of the existing data and policies regarding perinatal mental disorders in women are centered on antenatal depression during pregnancy or postnatal depression after childbirth. Few studies have paid attention to working women or investigated perinatal depressive symptoms during pregnancy, childbirth, and return to the workplace among employed women. Working mothers are especially vulnerable to workplace stressors because of sleep deprivation, the demands of caring for an infant, and overlooking psychological and emotional problems because of competing demands from home and work [13,14,15]. Perinatal depression, defined as depression in pregnancy, around childbirth, or within the first year postpartum, is a critical problem in households around the world and is often comorbid with other medical or mental health illnesses affecting all members of the family; it also often escapes detection and treatment [16,17]. If left untreated, maternal depression can disrupt the maternal–child bond; lead to suicide and/or infanticide; and increase workplace absenteeism, poor work performance, and disability costs for employers [18,19].

Work is very important for economic status and quality of life among women. However, studies have indicated that women’s work tends to be associated with a higher prevalence of job strain and low control over work conditions; therefore, women have higher odds of experiencing multiple disadvantages resulting in psychological distress compared to men in many industrialized countries [14,20]. There is substantial empirical evidence that employees (both men and women) who report job strain and a lack of decision latitude will experience increasing depressive symptoms over time [8,21]. Self-reported poor sleep has been associated with concurrent mood disturbances and an increased risk for future mood problems during pregnancy and the postpartum period [13,22], which warrants the screening of pregnant mothers for insomnia and depression. Little is known about the dynamic changes in the mental health status in pregnant employed women from pregnancy to returning to work, especially regarding depressive symptoms. There are few longitudinal data available on the period incidence or prevalence of perinatal depressive symptoms, especially in working women.

The aim of this study was to measure the period incidence or prevalence of perinatal depressive symptoms from pregnancy, childbirth, and return to the workplace in working women. Moreover, this study examined the association between sleep problems, perceived job stress, perceived support from others, and perinatal depressive symptoms during the perinatal period.

## 2. Methods

### 2.1. Research Setting and Recruitment

This was a longitudinal study of perinatal depressive symptoms among females employed in a large electronics manufacturer in Taiwan, conducted from August 2015 through October 2016. The research setting was Company C, a large electronics manufacturer with high labor-intensive employees (labor intensive industry refers to industries that require a substantial amount of human labor to produce industrial products) in Taiwan. This company has approximately 19,000 employees, and 41% were female. Company C was selected for the study setting upon consideration of company size and stability. The researchers inquired about the willingness of this company to participate in the study by first sending an explanatory letter about the research project and then visiting the company’s employee health management department director to explain the purpose of the research. Company C has many young female employees, and the employer and director of health management are interested in female employee health during pregnancy, and the negative impact of antenatal depression on occupational productivity and attendance. Thus, Company C was willing to provide assistance in administering this survey. Due to the low birth rate in Taiwan (total fertility rate in 2015 = 1.19 births per 1000 women), we planned to invite and include all eligible pregnant women in this company during the time of data collection (1 August 2015 to October 2016) and follow-ups were performed during their perinatal period. In Taiwan, the law stipulates that employers must provide 8 weeks of maternity leave for female employees.

This study was designed to follow participants during their perinatal period, and collect data of perinatal depressive symptoms in the following stages:

Stage 1: third trimester of pregnancy.

Stage 2: 3 weeks after childbirth.

Stage 3: 1 month after returning to work.

After obtaining consent from the employee health management department, occupational and environmental health nurses helped invite, distribute, and collect stage 1 questionnaires during the third trimester of pregnancy through regular prenatal health visits in the workplace. In this cohort study, participants were eligible if they were employed; worked full-time for at least 6 months during the pregnancy; were between 26 and 38 weeks of gestation at the time of data collection (August 2015 through October 2016); or had regular prenatal health visits during the third trimester of pregnancy. A total of 216 women were eligible for this study, as recorded by the human resources department. The study procedure, instruments, and all materials were reviewed and approved by the Institutional Review Board of E-Da Hospital in Taiwan. Written and signed informed consent was obtained from all participants.

### 2.2. Data Collection Instruments and Definitions

Of the 216 eligible women, 153 participated in the stage 1 survey. Follow-up survey of perinatal depressive symptoms of the 153 participators fixed cohort was conducted at 3 weeks after childbirth (stage 2), and at 1 month after returning to the workplace (stage 3). The study flow diagram and status of participation in the three stages are shown in Figure 1. The questionnaire content for stages 2 and 3 was similar to that for stage 1, being conducted by self-reported online questionnaire via email or telephone reminder. A total of 82 participants completed the 3 questionnaire stages; participation rate was 53.6% (82/153).

Questionnaires were used to collect data on demographics, personal lifestyle, pregnancy status, employment status, sleep status, perceived job strain and workplace support, family support, and the Edinburgh Postnatal Depression Scale (EPDS) scores. In stage 2, perceived job strain, workplace support, hours worked per day, and worksite may not be necessary when postpartum mothers are at home. Mean time required to complete the survey was 15 min. The details of the EPDS were described previously [23].

### 2.3. Demographics, Personal Lifestyle, and Employment Status

Level of education attained was used as a proxy measure for social class and categorized as: (1) high school or below, or (2) college or above. Monthly income was treated as individual work salary from this job and categorized into one of two grades: below USD 1250 dollars, and USD 1250 dollars and above per month. The participants were asked questions about their cigarette smoking behavior (current smoker and non-smoker); alcohol intake was limited to wine, hard alcohol and beer, and was categorized as no habit of alcohol consumption (or frequency of alcohol consumption was only once a week or less) and habit of alcohol consumption (frequency of alcohol consumption was more than once a week). The participants were office workers or clean-room workers (a room that is maintained virtually free of contaminants, used in laboratory work and in the production of precision parts for electronic equipment). Employment status was collected, including worksite (office vs. clean room), and work hours per day (<9 h a day or ≥9 h a day).

### 2.4. Perceived Job Strain and Social Support

We defined job strain as work demands that did not match the knowledge, skills, or abilities of the worker, and that challenged the woman’s ability to cope [24]. Participants responded to the following question: “Did you feel stress due to time pressures, amount of work, difficulty of the work, or empathy required in the last 3 months?” (yes/no). Social support was defined as interpersonal communication that provides psychosocial help for people in need. The most effective social support for mental health is related to family, friend, or spouse support. To classify these important types of support, participants were asked to respond to the following questions: “Were you able to obtain support from your family or friends?” (yes/no), and “Were you satisfied with the share of household responsibilities, childcare, or psychosocial help assumed by your partner?” (yes/no).

### 2.5. Pregnancy Status and Self-Reported Sleep Conditions

We collected data on the participants’ pregnancy status and self-reported sleep conditions. Participants were asked to respond to the following question: “Is this your first pregnancy?” (primiparas/multiparas). Sleep problems were surveyed and defined as difficulty falling asleep; difficulty remaining asleep (more than twice a week); the belief that one is not getting enough sleep, resulting in a disturbance of daily activities or normal social activities; and the use of medication for insomnia, focusing on symptoms during the preceding 4 weeks.

### 2.6. Family Function

The Family APGAR (Adaptability, Partnership, Growth, Affection, and Resolve) score was introduced in 1978 as a utilitarian screening instrument for family function and is used to assess a family member’s perception of family function and social support in family life by examining his/her satisfaction with family relationships [25]. The self-reported, five-item questionnaire is designed to detect dysfunction in the following areas: (1) family adaptation; (2) partnership; (3) growth; (4) affection; and (5) resolve. The instrument allows for three possible responses: almost always; some of the time; and hardly ever; and the scoring is 2, 1, and 0, respectively, for each of the five items in the questionnaire. Responses to the items are added, and thus the score can range from 0 to 10. A higher score indicates a greater degree of satisfaction with family function.

### 2.7. Edinburgh Postnatal Depression Scale

Antenatal depressive symptoms in this study were assessed using the EPDS. The EPDS was developed by Cox and Holden [26,27] to screen for the risk of postpartum depression. The EPDS scale was selected for this study because it is suitable for use during both pregnancy and the postpartum period [28]. It is a four-point self-reported scale composed of 10 items. Responses are scored from 0 to 3 and the total possible scores that can be obtained range from 0 to 30. Items 1, 2, and 4 are scored from 0 to 3, and items 3, 5, 6, 7, 8, 9, and 10 are scored in reverse order [28]. The Taiwanese version of the EPDS exhibits satisfactory sensitivity and specificity [29]. The previous study reported that the optimal cutoff points of the EPDS-T differ for detecting major depression during different trimesters: 13/14 for the second trimester and 12/13 for the third trimester [29]. In this study, we defined a score of 13 or more in the EPDS as perinatal depressive symptoms.

## 3. Statistical Analysis

The primary independent variables of interest were demographics, employment status, personal lifestyle, pregnancy status, self-reported sleep conditions, perceived job strain, and social support. The dependent variable was perinatal depressive symptoms using the EPDS during the three time-points. All analyses were performed using the Statistical Analysis System (SAS 9.3; SAS Institute, Cary, NC, USA) software. To compare the demographics and related factors between participants with and without follow-up during delivery and after returning to the workplace, univariate analyses were performed between groups with and without participation using the chi-square test or Student’s t-test. We adopted the EPDS scale instrument in this study and followed a fixed pregnancy employee’s cohort for three time-points. Hence, this study can estimate the prevalence and incidence of perinatal depressive symptoms during pregnancy, delivery, and after returning to the workplace among pregnant employees. Participant profiling was performed by comparing groups with and without antenatal depressive symptoms using the EPDS, and other independent variables using the chi-square test or Fisher’s exact test. A *p* value of less than 0.05 was considered statistically significant.

To determine whether sleep problems, perceived job strain, and perceived important support from others were associated with perinatal depressive symptoms in multivariate modeling, independent variables associated with perinatal depressive symptoms from pregnancy, delivery, to back to work during different perinatal stages in the multivariate modeling with *p* ≤ 0.05 in the bivariate logistic regression analysis were included in multiple logistic regression models. Estimated odds ratio (ORs) and 95% confidence interval (CIs) of the association were computed for the multiple logistic regressions. We examined goodness of fit for logistic regression models using the Hosmer–Lemeshow test, and the results revealed that the models had a good fit (*p*-value > 0.1).

## 4. Results

A total of 153 participants engaged in stage 1 of the study (Table 1). The mean age of the participants was 32.6 ± 3.18 years (Table 1). Most of the participants (62.1%) were between 25 and 33 years of age, and 79.1% had a college education. Among participants, 69.3% were office workers (not clean-room workers), and 62.1% of the participants averaged at least 9 h of work per day; mean number of hours of work per day was 9.5 ± 1.3 h. Most of the participants (99.3%) reported no smoking habit, and 98.7% reported no alcohol drinking habit. The percentage of participants with antenatal depressive symptoms using the EPDS scales (score ≥ 13) was 13.7%. A comparison of demographic information and some of the variables in the study revealed that participants of stage 2 were older than the non-participants (33.0 vs. 31.6, *p* = 0.0168). No differences in the other characteristics between the participants and non-participants were detected in stages 2 and 3.

The period prevalence of participants with perinatal depressive symptoms according to the EPDS scales (score ≥ 13) for the 3 stages was 13.7%, 16.8%, and 15.9%, respectively (Table 2). The mean EPDS score for each stage gradually increased from 14.6 to 15.1 to 15.3. The period incidence of perinatal depressive symptoms at 3 weeks after childbirth and at 1 month after returning to the workplace was 11% and 6.8%, respectively. At 3 weeks after childbirth period (stage 2), the incidence of perinatal depressive symptoms was 11.3% among primipara mothers and 9.3% among multiparas mothers. Conversely, at 1 month after returning to the workplace (stage 3) the incidence of perinatal depressive symptoms was 2.7% among primipara mothers and 8.9% among multipara mothers.

After excluding participants with only partial participation, a total of 82 participants completed the 3 surveys, including the baseline survey during the third trimester of pregnancy and follow-ups (3 weeks after childbirth and 1 month after returning to the workplace). Figure 2 shows the prevalence of perinatal depressive symptoms of this cohort (*n* = 82). Among the three time-points surveyed, the highest prevalence of perinatal depressive symptoms (18.1%) occurred 3 weeks after childbirth. Second, the prevalence of perinatal depressive symptoms at 1 month after returning to the workplace was 15.7%. During the entire perinatal period, 7.2% of the participants experienced constant feelings of depressive symptoms.

Perinatal depressive symptoms in the third trimester of pregnancy (EPDS ≥ 13) were associated with: sleep problems (*p* < 0.0001); perceived job strain (*p* = 0.0003); lack of support from family or friends (*p* = 0.0130); and lack of support from spouse (*p* = 0.0326; Table 3). Perinatal depressive symptoms at 3 weeks after childbirth (EPDS ≥ 13) were associated with: advanced maternal age (*p* = 0.0419); sleep problems (*p* = 0.0004); and lack of support from family or friends (*p* < 0.0001; Table 3). Perinatal depressive symptoms at 1 month after returning to the workplace (EPDS ≥ 13) were associated with: education level (*p* = 0.0406); monthly income (*p* = 0.0144); sleep problems (*p =* 0.0010); perceived job strain (*p* = 0.0003); lack of support from family or friends (*p* = 0.0428); and lack of support from spouse (*p* < 0.0001; Table 3).

The effects of sleep problems, perceived job strain, and important support from others on the three stages of perinatal depressive status were further evaluated using multiple logistic regressions (Table 4). During the third trimester of pregnancy, sleep problems (OR = 6.2, 95% CI = 2.1–19.3), perceived job stress (OR = 4.4, 95% CI = 1.5–14.3), and lack of support from family or friends (OR = 7.0, 95% CI = 1.3–40.8) were significantly associated with perinatal depressive symptoms. Sleep problems (OR = 6.0, 95% CI = 1.7–23.5) and lack of support from family or friends (OR = 27.6, 95% CI = 4.1–322.3) were associated with an increased risk of perinatal depressive symptoms at 3 weeks after childbirth. After returning to the workplace, the only significant risk factor for perinatal depressive symptoms was perceived job strain (OR = 18.2, 95% CI = 2.2–435.7) after controlling for all covariates (education, monthly income, sleep problems, and lack of support from family or friends and spouse) in the multiple regression model.

## 5. Discussion

To our knowledge, this is the first longitudinal study exploring perinatal depressive symptoms—from late pregnancy, delivery, and return to work—which reports the period prevalence and incidence of perinatal depressive symptoms among pregnant working women in Asia. The importance of identifying risk factors for perinatal depressive symptoms among pregnant employees is evident. Perinatal depression in this study was assessed using the EPDS, and the prevalence of perinatal depressive symptoms was 13.7% during the third trimester of pregnancy, 16.8% at 3 weeks after childbirth, and 15.9% at 1 month after returning to the workplace. The incidence of postpartum depressive symptoms was 11% at 3 weeks after childbirth and 6.8% at 1 month after returning to the workplace. We followed the 82-cohort pregnant women who completed all three time-points, a total of 7.2% of participants reported depressive symptoms from pregnancy, delivery, to their return to the workplace. We found that: (1) during the third trimester of pregnancy, sleep problems, perceived job strain, and lack of support from family or friends were significant risk factors; (2) sleep problems and lack of support from family or friends were associated with an increased risk of perinatal depressive symptoms at 3 weeks after childbirth; and (3) after returning to the workplace, perceived job strain was a significant risk factor.

In the present study, sleep problems in the third trimester of pregnancy (OR = 6.2) and sleep problems at 3 weeks after childbirth (OR = 6.0) were associated with perinatal depressive symptoms. The presence of insomnia or sleep disturbances during the perinatal period is a risk factor for depressive symptoms. There is strong evidence that self-reported poor sleep quality is associated with poor mood and might be a risk factor for mood problems during the perinatal period [21]. This bidirectional and additive relationship needs more clinical attention, as both sleep disturbance and depression are noted risk factors for adverse pregnancy outcomes [30]. Regarding sleep disturbances and their relationship to the prevalence of depression, sleep disturbances could either be a contributor to depression or a symptom of depression. This study has no way of clearly knowing what sleep disturbances represent in pregnant and postpartum women. Objective or self-reported healthcare assessments of sleep duration and quality are not included in the prenatal exams in Taiwan. Because symptoms of insomnia and poor sleep quality are independently associated with greater depressive symptoms across pregnancy and throughout the postpartum period, we suggest that objective or self-reported sleep checks should be included in the scope of prenatal examinations in the future.

Work is very important for economic status and quality of life among women, but a permanent job poses both risks and benefits to health. The results of the present study revealed that perceived job strain among women, during the third trimester of pregnancy (OR = 4.4) or 1 month after returning to work (OR = 18.2), was a significant risk factor for perinatal depressive symptoms. A Swedish longitudinal occupational health survey demonstrated that surviving a layoff is significantly associated with subsequent major depression in women, but not in men [31]. One study indicated that working mothers are especially vulnerable to workplace stressors because of sleep deprivation, demands of caring for an infant, and inability to engage in health promoting activities because of competing demands from home and work. Hence, workplace and parenting stressors place working mothers at greater risk for postpartum depression [32]. Identification of perinatal distress by midwives and other healthcare professionals is important, since distress may be linked to women’s complaints of fatigue [33]. In the present study most participants (around 79%) had higher education, and working mothers often worked more than the legally mandated 8 h (stage 1: 62.1% and stage 3: 59.0%), with the mean hours of work per day being 9.5 h. Moreover, nearly 30% participants were clean-room workers (stage 1: 30.7% and stage 3: 28.9%), who work 12-h shifts. Office workers had higher educational and compensation levels than clean-room workers, and generally worked about 8 h a day, but their positions encompass specific job responsibilities. Based on these findings, the working mothers in our study had a heavy work burden, and employment stress was negatively associated with women’s health after childbirth.

In this study, lack of support from family or friends was associated with increased perinatal depressive symptoms, especially in the third trimester of pregnancy (OR = 7.0) and at 3 weeks after childbirth (OR = 27.6). Women who are dissatisfied in their family or partner relationship were found to be four times more likely to experience perinatal distress in an Icelandic study [34]. Previous studies reported that poor social support [35], poor emotional support, and social isolation [36] are factors associated with antenatal depression. Adequate social support is vital for those most vulnerable to postpartum mood disorders, and receiving social support from others who share similar experiences may enhance the positive parenting experiences of mothers, which in turn can improve the psychosocial well-being of the mothers, strengthen the mother–child bond, and enhance the overall family dynamics for mothers and infants [37,38]. One study identified lack of workplace social support as a modifiable risk factor for postpartum anxiety, and suggested that increased social support provided by coworkers, supervisors, and the organization is associated with reduced odds of anxiety symptoms [39]. One study reported that perceived workplace and family support does not have moderating effects, and that stress management programs to decrease the levels of job strain in the workplace should be developed [22]. In Taiwan, the law stipulates that employers must provide 8 weeks of maternity leave for female employees. Employers and occupational health nurses should consider the implementation of stress management strategies, such as workload adjustments and stress detection awareness, which are likely to benefit female employees after childbirth by improving their mood and enhancing quality of life. This study suggested that perinatal education programs in the workplace should be provided, including the sharing of practical advice based on the experience of a mentor or coworker.

A study [29] mentioned that significant changes in physiological and psychological functions occur during each trimester of pregnancy, and that these changes influence factors such as leisure time and work-related physical activities, mood and anxiety states, quality and quantity of sleep, thyroid function, serum steroid hormone levels and hypothalamic–pituitary–adrenal axis function, essential fatty acid levels, hemostasis, and immune function. These factors could predispose women to the development of depression and might influence the incidence and phenomenology of depression during pregnancy. We followed the 82 cohort pregnant women who completed all three time-points, and discovered that 82 participants had a higher rate of perinatal depressive symptoms score (18.1%) in stage 2 (3 weeks after childbirth) than in the other two stages (third trimester of pregnancy and 1 month after returning to the workplace). Screening for postpartum depression has become more common, but postpartum mental health is largely ignored by maternity care providers [40]. When EPDS was first introduced, an initial application around the sixth postpartum week was recommended [19]. In this study, the second assessment was performed around the third postpartum week, as this is close to the postpartum blues period. Postpartum blues, also known as the baby blues or maternity blues, is a common self-limiting condition that begins shortly after childbirth and can present with a variety of symptoms, such as mood swings, irritability, and tearfulness [41]. Mothers may experience negative mood symptoms mixed with intense periods of joy. Many women feel confused about struggling with sadness after the joyous event of adding a new baby to the family, and often refuse to talk about their symptoms. However, talking about these emotions, changes, and challenges is one of the best ways to cope with the baby blues. The lack of accurate and timely diagnoses or attention to physical and mental disorders, specifically after birth, may result in irrecoverable emotional and cognitive impairment for women and their neonates [42]. This study suggested that paying attention to the key symptoms of maternity blues and women’s complaints, as well as implementing advance educational programs in the workplace, are essential. Moreover, this study revealed that the incidence of depression was higher in multiparous women compared with primiparous in the period after return to work (8.9% vs. 2.7%). Perhaps multiparous women have more time, work, and family conflicts and strains compared with primiparous in the period after return to work. It is also about actively seeking to provide support or assisting in providing successful coping experiences and offering education programs such as long-term midwife-led breastfeeding support and psychosocial support for employed women.

It is worth mentioning that a culturally sanctioned ritual of maternal rest and recovery with newborn care assistance, or “sitting (doing) the month,” has been traditionally practiced with specific diets, behavioral adjustment, and sufficient rest during the first postpartum month in Taiwan [43,44]. Several studies in Taiwan revealed that this culture can assist postpartum women to improve their physical condition, enhance their skills in caring for newborns, and provide valuable social support for Taiwanese postpartum women by adhering to postpartum recuperation. It seems that postpartum recuperation with newborn care assistance could reduce postpartum depressive symptoms [6,43,44,45].

The present study had some limitations. Firstly, assessment of risk factors was dichotomized, which was simplistic, and risk factor measurements mainly relied on self-report, which might have biased the results. Secondly, selection bias due to non-participant was inevitable. Third, the fact that standard instruments such as the concept of perceived job strain and social support were not applied in the present study could have led to a misclassification of information, may have affected the validity and reliability of the questionnaire, and may have caused a potential lack of comparability to the results of prior studies that used standard instruments. Due to the low birth rate in Taiwan, we invited and included all eligible pregnant women at Company C during the time of data collection. We did not perform sample size calculations before starting the study. However, this study has a large drop-out rate (46.4%; as 153 employees agreed to participate, but only 82 employees completed the 3 stages). The small sample size reduced the ability to detect modest but potentially meaningful and statistically significant associations, and the large drop-out rate may have reduced the power of the analysis. The confidence intervals of several ORs in this study were large, thus reflecting the imprecise estimation due to the small number of participants in the group. Therefore, some such associations might not have been detected as statistically significant. Returning to work while still breastfeeding presents a challenge. The reason is that the working mother who faces employment-related barriers to breastfeeding is, in essence, confronted with a conflict between her work and family roles. In this study we did not evaluate the issue of women’s intention to breastfeed, and whether that may have contributed to workplace stress, because the purpose of this study still focused on investigating the period of incidence of perinatal depressive symptoms and explored the impact of sleep problems, perceived job stress, and perceived support from others on perinatal depressive symptoms during the perinatal period. In order to have a full picture we suggest that future studies simultaneously evaluate women’s intention to breastfeed and workplace stress.

## 6. Conclusions

The present study, sleep problems, perceived job strain, and lack of support from family or friends among pregnant employees were significantly associated with perinatal depressive symptoms after adjusting for other factors. In view of the small sample size and the inclusion of employees of only one company in Taiwan, more studies to clarify the association between sleep problems, perceived job strain, and lack of support from family or friends among pregnant employees, and perinatal depressive symptoms will be worthwhile.

## Figures and Tables

**Figure 1 ijerph-20-03354-f001:**
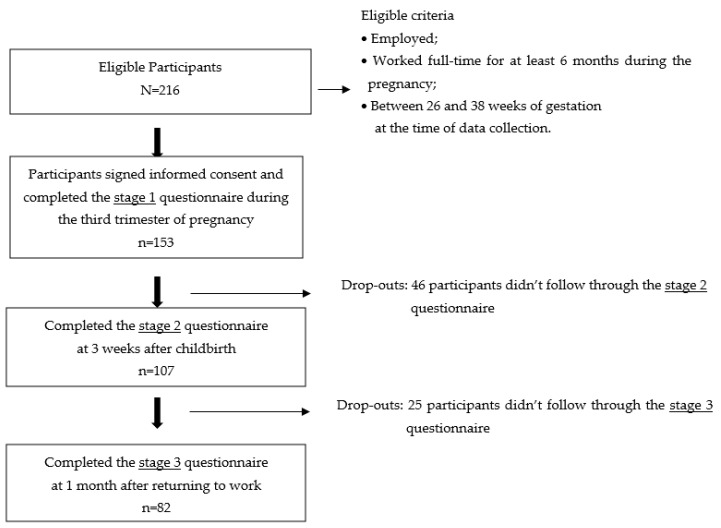
The study flow diagram.

**Figure 2 ijerph-20-03354-f002:**
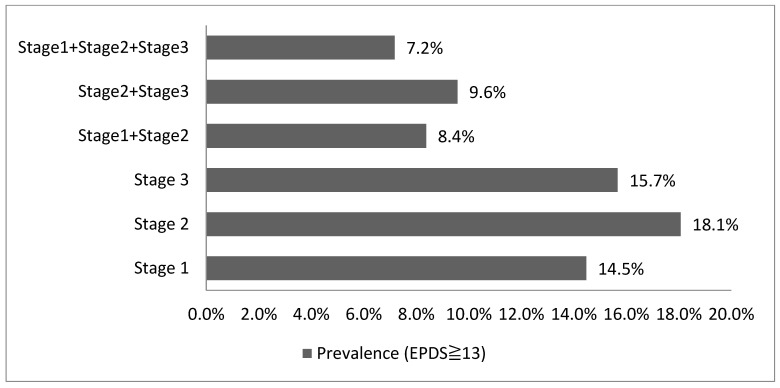
The cohort (*n* = 82) of prevalence trends of perinatal depressive symptoms in third trimester of pregnancy, at 3 weeks after childbirth, and at 1 month after returning to work.

**Table 1 ijerph-20-03354-t001:** Descriptive baseline characteristics of participants in the third trimester of pregnancy, at 3 weeks after childbirth, and at 1 month after returning to work.

Variable	Stage 1	Stage2			Stage3		
	Total Participants (*n* = 153)	Participants (*n* = 107)	Non-Participants (*n* = 46)	*p*-Value	Participants (*n* = 82)	Non-Participants (*n* = 25)	*p*-Value
Age (mean ± sd)	32.60 ± 3.18	33.0 ± 3.0	31.6 ± 3.3	0.0168	32.8 ± 3.0	32.3 ± 3.2	0.2961
Education level: College and above (%)	79.1	79.4	78.3	0.8695	83.1	74.3	0.1801
Monthly income: 1250 dollars and below (%)	66.0	64.5	69.6	0.5430	61.5	71.4	0.1940
Smoking: No (%)	99.3	100	97.8	0.1260	100	98.6	0.2746
Alcohol: No (%)	98.7	98.1	100	0.3506	97.6	100	0.1911
Hours worked per day: ≥9 (%)	62.1	--	--	--	59.0	65.7	0.3963
Worksite: Cleanroom (%)	30.7	--	--	--	28.9	32.8	0.5985
Baseline Family APGAR Score (mean ± sd)	8.3 ± 2.0	8.4 ± 1.9	8.0 ± 2.3	0.2831	8.3 ± 1.8	8.1 ± 2.3	0.5080
Baseline EPDS score ≥13 (%)	13.7	14.9	10.8	0.5009	14.4	12.8	0.7744

-- Analysis not required.

**Table 2 ijerph-20-03354-t002:** Period prevalence and incidence of perinatal depressive symptoms in the third trimester of pregnancy, at 3 weeks after childbirth, and at 1 month after returning to work.

	Perinatal Depressive Symptoms (EPDS Score ≥13)		
	Stage 1: Third Trimester of Pregnancy	Stage 2: 3 Weeks after Childbirth	Stage 3: 1 Month after Return to the Workplace
Total participants	N = 153	N = 107	N = 82
Depressed participants (Prevalence, %)	21 (13.7%)	18 (16.8%)	13 (15.9%)
EPDS mean score (range)	14.6 (13–18)	15.1 (13–24)	15.3 (13–19)
Incidence (%)	--	11.0% (11/100)	6.8% (5/74)
Primiparas (%) Multiparas (%)	-- --	6/53 = 11.3% 5/54 = 9.3%	1/37 = 2.7% 4/45 = 8.9%

-- Analysis not required.

**Table 3 ijerph-20-03354-t003:** Pregnancy status, self-reported mental and physical conditions, and perceived job strain and workplace support among 153 pregnant employee study participants and the Edinburgh Postnatal Depression Scale (EPDS) score.

	Stage 1: Third Trimester of Pregnancy			Stage 2: 3 Weeks after Childbirth			Stage 3: 1 Month after Returning to the Workplace		
Variables	EPDS		*p*-Value for X^2^ Test	EPDS		*p*-Value for X^2^ Test	EPDS		*p*-Value for X^2^ Test
	≥13 (*n* = 21)	<13 (*n* = 132)		≥13 (*n* = 18)	<13 (*n* = 89)		≥13 (*n* = 13)	<13 (*n* = 69)	
Age: ≥34 (advanced maternal age)	6 (28.6%)	52 (39.4%)	0.3424	4 (22.2%)	43 (48.3%)	0.0419	8 (61.5%)	26 (37.7%)	0.1092
Education level: college and above	14 (66.7%)	107 (81.0%)	0.1512§	12 (66.7%)	73 (82.0%)	0.1976 §	8 (61.5%)	60 (87.0%)	0.0406 §
Monthly income: 1250 dollars and below	17 (80.9%)	84 (63.6%)	0.1197	13 (72.2%)	56 (62.9%)	0.4520	12 (92.3%)	39 (56.5%)	0.0144 §
Current smoking	0 (0%)	1 (0.8%)	0.8627§	0 (0%)	0 (0%)	1.000 §	0 (0%)	0 (0%)	1.000 §
Alcohol intake	0 (0%)	2 (1.5%)	0.7436§	0 (0%)	2 (2.3%)	0.6905 §	0 (0%)	2 (2.9%)	0.7064 §
Primiparas	10 (47.6%)	75 (56.8%)	0.4307	10 (55.6%)	44 (49.4%)	0.6359	7 (53.9%)	33 (47.8%)	0.6904
Hours worked per day: ≥9	13 (61.9%)	82 (62.1%)	0.9848	--	--	--	9 (69.2%)	36 (52.2%)	0.2569
Worksite: cleanroom	13 (61.9%)	93 (70.5%)	0.4302	--	--	--	7 (53.9%)	48 (70.0%)	0.2686
Sleep problems	14 (66.7%)	23 (17.4%)	<0.0001	13 (72.2%)	25 (28.1%)	0.0004	10 (76.9%)	18 (26.1%)	0.0010 §
Perceived job strain	13 (61.9%)	31 (23.5%)	0.0003	--	--	--	12 (92.3%)	26 (37.7%)	0.0003
Lack of support from family or friends	4 (19.1%)	4 (3.0%)	0.0130 §	6 (33.3%)	2 (2.3%)	<0.0001 §	3 (23.1%)	3 (4.3%)	0.0428 §
Lack of support from spouse	2 (9.5%)	2 (1.5%)	0.0326 §	1 (5.6%)	1 (1.1%)	0.3095 §	5 (38.5%)	0 (0%)	<0.0001 §

§ Fisher’s exact test; -- Analysis not required.

**Table 4 ijerph-20-03354-t004:** Multiple logistic regressions of perinatal depressive symptoms (EPDS ≥ 13) during the three stages for risk factors including sleep problems, perceived job strain, and important support from others among participants, adjusted for other variables.

	Stage 1: * Third Trimester of Pregnancy (*n* = 153)			Stage 2: ** 3 Weeks after Childbirth (*n* = 107)			Stage 3: *** 1 Month after Return to the Workplace (*n* = 82)		
Variables	OR	95% CI	*p*-Value	OR	95%CI	*p*-Value	OR	95%CI	*p*-Value
Sleep problems	6.2	2.1–19.3	0.0010	6.0	1.7–23.5	0.0057	--	--	--
Perceived job strain	4.4	1.5–14.3	0.0091				18.2	2.2–435.7	0.0187
Lack of support from family or friends	7.0	1.3–40.8	0.0244	27.6	4.1–322.3	0.0021	--	--	--
Lack of support from spouse	--	--	--	--	--	--	--	--	--
Hosmer–Lemeshow test	0.3568			0.8445			0.4620		

* In this multiple logistic regress, we examined associated factors including sleep problems, perceived job strain, lack of support from family or friends and spouse. ** In this multiple logistic regression, we examined associated factors including age, sleep problems and lack of support from family or friends. *** In this multiple logistic regression, we examined associated factors including education level, monthly income, sleep problems, perceived job strain, and lack of support from family or friends and spouse. --Not significant.

## Data Availability

The data presented in this study are available on request from the corresponding author.

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
