# Peer review of "Perinatal Depressive Symptoms among Pregnant Employees in Taiwan"

_ijerph, 2023, doi:10.3390/ijerph20043354_

Round 1

Reviewer 1 Report

It was my pleasure to review this manuscript dealing with perinatal depressive disorders among pregnant employees in Taiwan.

I found the topic quite interesting. But with the sole objective of improving the quality of the manuscript, I will allow myself to make a few comments:

Introduction: At the end of this section, the objective of this study is stated. I think that the objective should be written in a clearer and more concrete way. I am not sure what the authors mean by investigating the period of incidence of perinatal depressive symptoms during pregnancy, childbirth and return to the workplace. They then say that the impact of sleep problems, perceived work stress, and support received from others on depressive symptoms of pregnancy during the three periods described above will be explored. After so many things, it was not clear to me what the purpose of the study is.

Methods: It is not specified how the sample size was calculated to be representative. A margin of error of 5% and a confidence interval of 95% are normally accepted. The number of individuals that made up the study population was not expressed, nor was the minimum number of the necessary sample meeting the criteria I listed above.

At the end, a total of 82 individuals completed the study. Is this sample really representative? It seems too small to me for the number of people who work in this company.

Nor is it defined how the sample was chosen. Was it done randomly, or just for convenience?

Conclusions: It would be convenient to make a section of conclusions independently from the discussion. The conclusions must support the results and respond to the objectives that were raised at the beginning. Actually the conclusions do not meet these criteria.

References: It seems that the authors wanted to use the Vancouver format, but it was not used correctly. I suggest reviewing the rules on bibliographic references accepted by this Journal and writing them correctly.

Kind regards

Author Response

Comments and Suggestions for Authors:

Reviewer 1

It was my pleasure to review this manuscript dealing with perinatal depressive disorders among pregnant employees in Taiwan. I found the topic quite interesting. But with the sole objective of improving the quality of the manuscript, I will allow myself to make a few comments:

Introduction: At the end of this section, the objective of this study is stated. I think that the objective should be written in a clearer and more concrete way. I am not sure what the authors mean by investigating the period of incidence of perinatal depressive symptoms during pregnancy, childbirth, and return to the workplace. They then say that the impact of sleep problems, perceived work stress, and support received from others on depressive symptoms of pregnancy during the three periods described above will be explored. After so many things, it was not clear to me what the purpose of the study is.

Response: Thank you for pointing this out. In Taiwan, little is known about the dynamic changes in the mental health status of employed pregnant women. This study conducted a longitudinal study to investigate the period incidence or prevalence of perinatal depressive symptoms from pregnancy, childbirth, and return to the workplace in working women. Moreover, we explored the impact of sleep problems, perceived job stress, and perceived support from others on perinatal depressive symptoms during the perinatal period. Such information may provide insight for employers or occupational and environmental health nurses about the early detection of symptoms and appropriate healthcare referrals for improving pregnant employees’ quality of life or preventing major depression. We have revised it clearer and more concrete way. Please see page 2, paragraph 3 (red color).

Methods: It is not specified how the sample size was calculated to be representative. A margin of error of 5% and a confidence interval of 95% is normally accepted. The number of individuals that made up the study population was not expressed, nor was the minimum number of the necessary sample meeting the criteria I listed above.

Response: Thank you for pointing this out. Due to the low birth rate in Taiwan (total fertility rate in 2015=1.19 births per 1000 women), we invited and included all eligible pregnant women at Company C during the time of data collection (August 1, 2015, to October 2016) and follow-ups were performed during their perinatal period. As such, we did not perform sample size calculations before starting the study. The small sample size reduced the ability to detect modest but potentially meaningful and statistically significant associations, and the large drop-out rate may have reduced the power of the analysis. The confidence intervals of several ORs in this study were large, thus reflecting the imprecise estimation due to the small number of participants in the group. Therefore, some such associations might not have been detected as statistically significant. In view of the small sample size and the inclusion of employees of one company in Taiwan, more studies to clarify the association between sleep problems, perceived job strain, and lack of support from family or friends among pregnant employees, and perinatal depressive symptoms are worthwhile. We have added these explanations and suggestions to study limitations. Please see page 11 (red color).

In the end, a total of 82 individuals completed the study. Is this sample really representative? It seems too small to me for the number of people who work in this company.

Nor is it defined how the sample was chosen. Was it done randomly, or just for convenience?

Response: Thank you for pointing this out. Due to the low birth rate in Taiwan, we invited and included all eligible pregnant women at Company C during the time of data collection (August 1, 2015, to October 2016) and follow-ups were performed during their perinatal period. In this cohort study, participants were eligible if they were employed; worked full-time for at least 6 months during the pregnancy; were between 26 and 38 weeks of gestation at the time of data collection; had regular prenatal health visits during the third trimester of pregnancy. A total of 216 women were eligible for this study, as recorded by the human resources department. Of the 216 eligible women, 153 participated in the stage 1 survey. A total of 82 individuals completed three questionnaire stages. The information has been addressed in the method (red color).

We agreed with the reviewer’s opinion. A small sample size affected the power and was not representative. We added this in the limitation part. In view of the small sample size and the inclusion of employees of one company in Taiwan, more studies to clarify the association between sleep problems, perceived job strain, and lack of support from family or friends among pregnant employees, and perinatal depressive symptoms are worthwhile. We have added these explanations and suggestions to study limitations. Please see page 11 (red color).

Conclusions: It would be convenient to make a section of conclusions independently from the discussion. The conclusions must support the results and respond to the objectives that were raised at the beginning. Actually, the conclusions do not meet these criteria.

Response: Thank you for pointing this out. The conclusions regarding the results and the objectives of this study were raised at the beginning of the discussion part. Please see page 8 (red color).

References: It seems that the authors wanted to use the Vancouver format, but it was not used correctly. I suggest reviewing the rules on bibliographic references accepted by this Journal and writing them correctly.

Response: Thank you for pointing this out. The author has revised the references according to the reviewer’s suggestion.

Reviewer 2 Report

 Perinatal depressive disorders among pregnant employees in Taiwan

This is a longitudinal study of prevalence of depression in pregnant women in the third trimester, the period shortly after giving birth, and shortly after returning to work. Investigators sought to identify factors associated with depression and found that sleep problems, perceived job strain, and lack of support from family or friends among pregnant employees were significantly associated with perinatal depressive symptoms after adjusting for other factors.

One question that was not addressed was what was the average time period between giving birth and employees’ return to work? Along the same lines, what are governmental or workplace policies about the length of time off after giving birth that are afforded to women? Does your study have any policy implications in this area?

Another issue not evaluated nor discussed is the issue of women’s intention to breastfeed, and whether that may have contributed to workplace stress. Can the author comment on breastfeeding rates in Taiwan and whether these differ between working and non-working mothers? Also, availability of child care might be mentioned. How big of a contributing stress is this to working women and their families? Do workplaces offer childcare? One workplace intervention that could potentially lower depression in breastfeeding women is to ensure they have time to pump breastmilk during working hours.

Regarding sleep disturbances and their relationship to prevalence of depression: It would be worthwhile to comment that sleep disturbances could either be a contributor to depression or a symptom of depression, and investigators have no way of clearly knowing what sleep disturbances represent in the pregnant and postpartum woman.

The authors suggest that educational programs in the workplace might be helpful. Can you give an idea of what these programs might include, and how employees might access them?

Specific comments:

Line 27: “Women are much more likely to be diagnosed with depression…” Comment:  compared to who or what?

Line 43: “lead to suicide and/OR infanticide”

Paragraph starting with Line 45: would reorganize so that sentence starting on Line 54 about job strain and the sentence following it come after first sentence in paragraph about job strain.

Line 124: Did participants “refuse” to complete surveys or just not follow through? I’m guessing the latter. “Refuse” indicates some objection while my sense is that participants were likely just overwhelmed with other time-consuming responsibilities.

Lines 221-222: I would combine the first two sentences: “A total of 153 participants participated in stage 1 of the study.”

Table 2: An interesting finding that did not receive much comment was the finding that rates of depression were higher in multiparous women compared with primiparous in the period after return to work (8.9% vs. 2.7%). This merits a comment from the authors in the Discussion section.

Line 299: Investigators should include comparative rates of depression in pregnant and postpartum women from around the world for reference and context. Is there anything unusual about rates of depression in working women in Taiwan compared with women in other parts of the world?

Line 377: Is postpartum depression screening recommended/mandated by health agencies in Taiwan as it is in the US? Even so, mandates are for healthcare practitioners, not workplace supervisors or employee health nurses. It would seem to be an overreach to have routine screening for postpartum depression in workplaces. I think that recommendations for intervention should also be referred back to healthcare providers (ask women about their plans to return to work, whether they have enough support, etc.) in addition to considering workplace interventions.

Author Response

Comments and Suggestions for Authors:

Reviewer 2

This is a longitudinal study of the prevalence of depression in pregnant women in the third trimester, the period shortly after giving birth, and shortly after returning to work. Investigators sought to identify factors associated with depression and found that sleep problems, perceived job strain, and lack of support from family or friends among pregnant employees were significantly associated with perinatal depressive symptoms after adjusting for other factors.

One question that was not addressed was what was the average time period between giving birth and employees’ return to work. Along the same lines, what are governmental or workplace policies about the length of time off after giving birth that is afforded to women? Does your study have any policy implications in this area?

Response: Thank you for pointing this out. In Taiwan, the law stipulates that employers must provide 8 weeks of maternity leave for female employees. Hence, the average time period between giving birth and employees’ return to work was 8 weeks. In this study, we suggested employers and occupational health nurses should consider the implementation of stress management strategies, such as workload adjustments and stress detection awareness, which are likely to benefit female employees after childbirth by improving their mood and enhancing their quality of life. Moreover, this study suggested that perinatal education programs in the workplace should be provided, including the sharing of practical advice based on the experience of a mentor or coworker. These contents have been addressed in the manuscript. Please see page 10, paragraph 1 (red color).

Another issue not evaluated nor discussed is the issue of women’s intention to breastfeed, and whether that may have contributed to workplace stress. Can the author comment on breastfeeding rates in Taiwan and whether these differ between working and non-working mothers? Also, availability of child care might be mentioned. How big of a contributing stress is this to working women and their families? Do workplaces offer childcare? One workplace intervention that could potentially lower depression in breastfeeding women is to ensure they have time to pump breastmilk during working hours.

Response: We appreciate your suggestion. We agreed with the reviewer’s opinion about the issue of women’s intention to breastfeed and whether that may have contributed to workplace stress. In Taiwan, the law stipulates that employers must provide 8 weeks of maternity leave for female employees. The government encourages companies or industries to provide breastfeeding support services, such as breast-pumping breaks and lactation rooms. Employees must bring their own breast pumps. Returning to work while still breastfeeding presents a challenge. The reason is that the working mother who faces employment-related barriers to breastfeeding is, in essence, confronted with a conflict between her work and family roles. In fact, we are going to write another manuscript to discuss this important topic. Hence, the purpose of this study still focused on investigating the period of incidence of perinatal depressive symptoms during pregnancy, childbirth, and return to the workplace and explored the impact of sleep problems, perceived job stress, and perceived support from others on perinatal depressive symptoms during the perinatal period. 

Regarding sleep disturbances and their relationship to prevalence of depression: It would be worthwhile to comment that sleep disturbances could either be a contributor to depression or a symptom of depression, and investigators have no way of clearly knowing what sleep disturbances represent in the pregnant and postpartum woman.

Response: Thank you for pointing this out. We agreed with the reviewer’s opinion. We have added this comment to the revised manuscript. Please see page 9, paragraph 1 (red color).

The authors suggest that educational programs in the workplace might be helpful. Can you give an idea of what these programs might include, and how employees might access them?

Response: Thank you for pointing this out. This study suggested that perinatal education programs in the workplace should be provided, including the sharing of practical advice based on the experience of a mentor or coworker. Moreover, we suggested that the workplace can offer support and provide successful coping experiences such as long-term midwife-led breastfeeding support and psychosocial support for employed women. We have added these comments to the revised manuscript. Please see page 10, paragraph 1, and paragraph 2 (red color).

Specific comments:

Line 27: “Women are much more likely to be diagnosed with depression…” Comment:  compared to who or what?

Response: Thank you for pointing this out. Women are much more likely to be diagnosed with depression during the first year after childbirth as compared to other times. We have revised this sentence in the revised manuscript. Please see page 1, line 27(red color).

Line 43: “lead to suicide and/OR infanticide”

Response: Thank you for pointing this out. We have revised it in the revised manuscript. Please see page 2, paragraph 1 (red color).

Paragraph starting with Line 45: would reorganize so that sentence starting on Line 54 about job strain and the sentence following it come after first sentence in paragraph about job strain.

Response: Thank you for pointing this out. We have reorganized that sentence according to the reviewer’s suggestion. Please see page 2, paragraph 2.

Line 124: Did participants “refuse” to complete surveys or just not follow through? I’m guessing the latter. “Refuse” indicates some objection while my sense is that participants were likely just overwhelmed with other time-consuming responsibilities.

Response: Thank you for pointing this out. We agreed with your opinion about participants didn’t follow through. We have revised it according to the reviewer’s suggestion. Please see page 3 (red color).  

Lines 221-222: I would combine the first two sentences: “A total of 153 participants participated in stage 1 of the study.”

Response: Thank you for pointing this out. We have revised it according to the reviewer’s suggestion. Please see page 5 (red color).  

Table 2: An interesting finding that did not receive much comment was the finding that rates of depression were higher in multiparous women compared with primiparous in the period after return to work (8.9% vs. 2.7%). This merits a comment from the authors in the Discussion section.

Response: Thank you for pointing this out. This study revealed that the incidence of depression was higher in multiparous women compared with primiparous in the period after return to work (8.9% vs. 2.7%). Perhaps multiparous women have more time, work, and family conflicts and strains compared with primiparous in the period after return to work. We suggested that the workplace can offer support and provide successful coping experiences such as long-term midwife-led breastfeeding support and psychosocial support for employed women. We have added it to the discussion. Please see page 10, paragraph 2 (red color).

Line 299: Investigators should include comparative rates of depression in pregnant and postpartum women from around the world for reference and context. Is there anything unusual about rates of depression in working women in Taiwan compared with women in other parts of the world?

Response: Thank you for pointing this out. According to a World Health Organization survey (World Health Organization 2009), 10% to 15% of women in developed countries and 20% to 40% of women in developing countries experience depression during pregnancy or after childbirth. We have two reasons that we didn’t include comparative rates of depression in pregnant and postpartum women from around the world in this manuscript. First, this study paid attention to working women and investigated perinatal depressive symptoms during pregnancy, childbirth, and return to the workplace among employed women. Previous studies were restricted to full-time housewives or postpartum women from hospitals or postpartum nursing homes. Second, this study has a large drop-out rate. The large drop-out rate may have reduced the power of the analysis. In view of the small sample size and the inclusion of employees of one company in Taiwan, the result was not representative. We added these explanations to the study limitation. Please see page 11 (red color).

Line 377: Is postpartum depression screening recommended/mandated by health agencies in Taiwan as it is in the US? Even so, mandates are for healthcare practitioners, not workplace supervisors or employee health nurses. It would seem to be an overreach to have routine screening for postpartum depression in workplaces. I think that recommendations for intervention should also be referred back to healthcare providers (ask women about their plans to return to work, whether they have enough support, etc.) in addition to considering workplace interventions.

Response: Thank you for pointing this out. In Taiwan, we did not have regular antenatal and postpartum depression screening in the scope of prenatal examinations. Antenatal depression has been identified as a serious health problem, yet remains a neglected component of care for women in the third trimester of pregnancy. One reason for this may be the difficulty in diagnosing antenatal depression because some physiologic signs of pregnancy overlap with the symptoms of antenatal depression, and healthcare providers tend to focus mainly on the physical health aspects of pregnancy. Regarding postpartum depression, physicians who care for infants and children encounter mothers repeatedly, this may be an important chance that they can detect symptoms of postpartum depression. Most Taiwanese women continue to work throughout their pregnancy. Such implementation can be used to understand the current state of the workplace environment and facilitate the implementation of a supportive workplace climate by employers, supervisors, and occupational and environmental health nurses for pregnant employees, which may decrease antenatal depression or psychological distress and improve the health of pregnant employees.

Reviewer 3 Report

The longitudinal study carried out by the author investigates an important aspect of perinatal health in a sample of Taiwanese women employees. The results obtained should be further argued and are only partially acceptable.

In particular, the Introduction section contains numerous sentences that refer to a single bibliographic citation. These statements must therefore be enriched with further references drawing from the current rich international bibliography.

To foster greater reader understanding, a new section should be introduced regarding the current state of the art and scientific knowledge gained in the field of perinatal depression in Taiwanese women.

However, the part that needs to be radically revised is that relating to the EPDS cut-off used by the author of the manuscript. In fact, the researcher improperly used the EPDS cut-off value of 12/13 even in the postpartum period despite this value referring exclusively to the third trimester of pregnancy according to the validation of Suh and collaborators (2007). The current scientific literature suggests that the EPDS cut-off 9/10 should be used in postpartum Taiwanese women (Heh et al., 2001).

The adoption of the postpartum EPDS cut-off suggested by the current literature will probably redesign a the manuscript configuration and open up new scientific meanings.

Author Response

Comments and Suggestions for Authors:

Reviewer 3

The longitudinal study carried out by the author investigates an important aspect of perinatal health in a sample of Taiwanese women employees. The results obtained should be further argued and are only partially acceptable.

In particular, the Introduction section contains numerous sentences that refer to a single bibliographic citation. These statements must therefore be enriched with further references drawing from the current rich international bibliography.

Response: Thank you for pointing this out. We have added several new references to the introduction section according to the reviewer’s suggestions. Please see the Introduction section (pages 1-2).

To foster greater reader understanding, a new section should be introduced regarding the current state of the art and scientific knowledge gained in the field of perinatal depression in Taiwanese women.

Response: Thank you for your suggestion. It is worth mentioning that a culturally sanctioned ritual of maternal rest and recovery with newborn care assistance, or “sitting (doing) the month,” has been traditionally practiced with specific diets, behavioral adjustment, and sufficient rest during the first postpartum month in Taiwan [40,41]. Several studies in Taiwan revealed that this culture can assist postpartum women to improve their physical condition, enhance their skills in caring for newborns, and provide valuable social support for Taiwanese postpartum women by adhering to postpartum recuperation. It seems that postpartum recuperation with newborn care assistance could reduce postpartum depressive symptoms [40,41,42,43]. We added this information in the discussion part. Please see page 10, paragraph 3 (red color).

However, the part that needs to be radically revised is that relating to the EPDS cut-off used by the author of the manuscript. In fact, the researcher improperly used the EPDS cut-off value of 12/13 even in the postpartum period despite this value referring exclusively to the third trimester of pregnancy according to the validation of Suh and collaborators (2007). The current scientific literature suggests that the EPDS cut-off 9/10 should be used in postpartum Taiwanese women (Heh et al., 2001). The adoption of the postpartum EPDS cut-off suggested by the current literature will probably redesign the manuscript configuration and open up new scientific meanings.

Response: We appreciate your suggestion. EPDS cut-off value varies from different studies. We reviewed a similar, updated, prospective study in Taiwan (Teng et al., 2005*), and this study concluded that the Taiwanese version of EPDS-T had satisfactory sensitivity and better specificity than BDI-II, that the 12/13 cutoff point was the best for screening PPD. The previous study reported that the optimal cutoff points of the EPDS-T differ for detecting major depression during different trimesters: 13/14 for the second trimester and 12/13 for the third trimester (Su et al. 2007).

Hence, this study still adopted a score of 13 or more in the EPDS as perinatal depressive symptoms.

* Hui-Wen Teng , Chun-Sen Hsu, Shou-Mei Shih, Mong-Liang Lu, Jan-Jhy Pan, Winston W Shen. Screening postpartum depression with the Taiwanese version of the Edinburgh Postnatal Depression scale. Compr Psychiatry 2005;46(4):261-5.

Round 2

Reviewer 1 Report

Dear authors,

It was a pleasure to review this second improved version of this manuscript dealing with perinatal depressive disorders among pregnant employees in Taiwan.

The version is a bit better than the first, but I still have some comments to make.

In the introduction, he starts out by talking about the methodology, but he doesn't really do an introduction explaining how depression affects pregnant and postpartum women. There are also no depression incidence data both globally and locally in Taiwan.

In the first revision, I recommended that the authors clearly state the objective of the research in the last paragraph. I believe that this recommendation was not taken into account because I have not yet read what the objective of this study is.

For the rest, I believe that the other recommendations were incorporated and although the study has important limitations that were already included by the authors, I consider that this version improved compared to the previous one.

One point that seems important to me is to define how long it takes women after childbirth to return to work, since the authors say that phase three of the study coincides with the return to work, but they do not say how long that time is. I imagine that in each country it will be different. That is why it is important to specify it.

Kind regards

Author Response

Comments and suggestions

Reviewer 1:

In the introduction, he starts out by talking about the methodology, but he doesn't really do an introduction explaining how depression affects pregnant and postpartum women. There are also no depression incidence data both globally and locally in Taiwan.

Response: Thank you for pointing this out.

According to a World Health Organization survey [4], 10% to 15% of women in developed countries and 20% to 40% of women in developing countries experience depression during pregnancy or after childbirth. A longitudinal study [5] at different time points during pregnancy and in the postpartum period showed that the period prevalence depression was 12.4% in pregnancy and 9.6% in the postpartum period, and the cumulative incidence of depression in pregnancy and in the postpartum period was 2.2% and 6.8%, respectively. Another study in Taiwan [6] revealed that the prevalence of perinatal depression was 13.2%, and the incidence of depression was 4.7% (16th week) and 3.4% (28th week), and 5.1% during the postpartum period. Depression during pregnancy is an important health problem because depressive symptoms not only lead to serious psychologic problems in women but also have a significant effect on the family [4,5,6]. Postpartum depression threatens not only the health of mothers but also the health of infants, and infants of depressed mothers may have developmental disorders, reduced cognitive function, as well as problems pertaining to social communication with their parents and peers [7,8,9]. We added the contents to the revised manuscript. Please see page 1 (red color).

In the first revision, I recommended that the authors clearly state the objective of the research in the last paragraph. I believe that this recommendation was not taken into account because I have not yet read what the objective of this study is.

Response: Thank you for pointing this out. The aim of this study was to measure the period incidence or prevalence of perinatal depressive symptoms from pregnancy, childbirth, and return to the workplace in working women. Moreover, this study examined the association between sleep problems, perceived job stress, perceived support from others, and perinatal depressive symptoms during the perinatal period. We revised the objective of this study according to the reviewer’s suggestion. Please see page 2 (red color).

One point that seems important to me is to define how long it takes women after childbirth to return to work since the authors say that phase three of the study coincides with the return to work, but they do not say how long that time is. I imagine that in each country it will be different. That is why it is important to specify it.

Response: Thank you for pointing this out. In Taiwan, the law stipulates that employers must provide 8 weeks of maternity leave for female employees. Hence, the average time period between giving birth and employees’ return to work was 8 weeks. Stage 3 was conducted one month after returning to work. We added this information to the revised manuscript. Please see page 3 (red color).

Reviewer 3 Report

Now, I consider the work publishable on IJERPH. Consistently with what reported by the author of the manuscript in lines 75-76, however, I propose to adapt the title to "Perinatal depressive symptoms among females employed in Taiwan".

Author Response

Comments and suggestions

Reviewer 3:

Now, I consider the work publishable on IJERPH. Consistently with what reported by the author of the manuscript in lines 75-76, however, I propose to adapt the title to "Perinatal depressive symptoms among females employed in Taiwan".

Response: Thank you for your suggestion. We revised the title of the manuscript according to the reviewer’s suggestion. Please see page 1.